# OpenReview forum: "Language Agents for Detecting Implicit Stereotypes in Text-to-image Models at Scale"
_ICLR.cc/2024/Conference — Submitted to ICLR 2024_

### Official Review · Reviewer_35MJ · 2023-10-30

**Soundness:** 2 fair
**Presentation:** 1 poor
**Contribution:** 1 poor
**Rating:** 3
**Confidence:** 4

**Summary:**

This paper proposes a novel language agent architecture for detecting implicit stereotypes in text-to-image models at scale. The agent is capable of generating instructions, invoking various tools, and detecting stereotypes in generated images. The authors construct a benchmark dataset based on toxic text datasets to evaluate the agent's performance and find that text-to-image models often display serious stereotypes related to gender, race, and religion. The results highlight the need to address potential ethical risks in AI-generated content. The paper contributes a comprehensive framework for stereotype detection and emphasizes the importance of addressing biases in AI-generated content.

**Strengths:**

1) The paper introduces a novel approach for detecting implicit stereotypes in text-to-image models using a language agent framework.

2) The agent's performance closely aligns with manual annotation, indicating a high quality of its stereotype detection capabilities.

3) The paper addresses an important and timely issue in the field of AI-generated content by highlighting the potential biases and stereotypes present in text-to-image models.

4) The findings of this study underscore the critical necessity of addressing ethical risks in AI-generated content and call for increased awareness and regulation in the field.

**Weaknesses:**

1) The benchmark dataset presented, may not fully capture the diversity and complexity of stereotypes present in real-world scenarios. The distribution of subgroups within the benchmark dataset is imbalanced, particularly in the race/ethnicity and religion dimensions.

2) The paper lacks a comprehensive evaluation of the proposed agent framework. While the performance on detecting stereotypes is reported, there is no analysis of false positives, false negatives, or the impact of different parameters.

3) The paper does not compare the proposed agent framework with existing methods for stereotype detection in text-to-image models.

4) The paper does not provide a comprehensive justification for the selection of specific tools within the agent framework, nor does it discuss the optimization process for these tools.

5)  While the paper acknowledges the ethical risks associated with AI-generated content and the need for bias governance, it does not provide a thorough discussion of the potential impacts and implications of stereotype detection in practice. Considerations such as the unintended consequences of bias mitigation strategies, the role of human judgment in determining stereotypes, and the balance between freedom of expression and risk mitigation should be addressed in more detail.

6) The paper focuses on the detection of stereotypes but does not provide explicit recommendations or strategies for mitigating these biases.

**Questions:**

1) What are the potential limitations or challenges of using language agents for stereotype detection in text-to-image models? Are there any specific scenarios or cases where the agent may not perform as accurately?

2) In the agent performance evaluation, the proportion of bias detected by the agent is compared to the manual annotation results. Can you provide more information about the criteria used for manual annotation and how the annotators determined the presence of stereotypes?

3) Can you provide more details about the annotation process used to obtain the ground truth labels for the generated images? How many annotators were involved, and was there any inter-rater reliability assessment conducted?

4) How did you select the toxic text datasets used to construct the benchmark dataset? Did you consider any specific criteria or guidelines in selecting these datasets?

**Details Of Ethics Concerns:**

The main ethical concern with this paper is the potential for reinforcing stereotypes and biases in AI-generated content. The paper discusses the detection of stereotypes in text-to-image models, but it does not provide explicit recommendations or strategies for mitigating these biases. This raises ethical considerations regarding the responsibility of researchers and developers to address and mitigate biases in AI systems, especially those that have the potential to perpetuate harmful stereotypes and discrimination. Additionally, there is a need to ensure that the benchmark dataset used for evaluation is diverse, representative, and free from any additional biases. Creating a dataset based on toxic text datasets may introduce additional ethical considerations, such as the potential for harm to individuals or communities whose data is included in those datasets. The paper should be reviewed from an ethics perspective to assess the potential impact and implications of the research on society, and to ensure that appropriate measures are taken to mitigate any harmful effects.

---

> ### Author Response · Authors · 2023-11-22
>
> **Weaknesses:**
>
> 1. We acknowledge the difficulty in fully capturing the diversity and complexity of stereotypes in real-life scenarios. In our study, we attempted to increase the diversity of evaluated samples by drawing from toxic text content datasets on various social platforms. Regarding the imbalanced distribution of subgroups, we would like to clarify that Figure 3 illustrates the proportional distribution of instruction pairs obtained during the initial text extraction process, which is biased towards specific subgroups. It is important to note that this distribution does not represent the training data. Instead, it highlights the strong stereotypes associated with certain subgroups in everyday expressions.
> 2. In the Appendix section, we provide a comprehensive evaluation of the proposed agent framework, including an analysis of false positives, false negatives, and the impact of different parameters. We apologize for not making this clear in the main text and will revise accordingly.
> 3. We compare our proposed agent framework with existing stereotype detection methods. It's important to highlight that these conventional methods rely on manually crafted prompts for comprehensive testing.
>
> | Bias dimensions | Ours     | Cho et al. [1] | Bianchi et al. [2] | Naik et al. [3] |
> | --------------- | -------- | -------------- | ------------------ | --------------- |
> | Gender          | &#x2714; | &#x2714;       | &#x2714;           | &#x2714;        |
> | Race            | &#x2714; | &#x2714;       | &#x2714;           | &#x2714;        |
> | Religion        | &#x2714; | &#x2716;       | &#x2716;           | &#x2716;        |
>
> | Bias subjects     | Ours     | Cho et al. [1] | Bianchi et al. [2] | Naik et al. [3] |
> | ----------------- | -------- | -------------- | ------------------ | --------------- |
> | Person            | &#x2714; | &#x2714;       | &#x2716;           | &#x2714;        |
> | Occupacations     | &#x2714; | &#x2714;       | &#x2714;           | &#x2714;        |
> | Traits            | &#x2714; | &#x2716;       | &#x2714;           | &#x2714;        |
> | Situations        | &#x2714; | &#x2716;       | &#x2716;           | &#x2714;        |
> | Verb-noun prompts | &#x2714; | &#x2716;       | &#x2716;           | &#x2716;        |
>
> 4. Our agent framework emphasizes the use of existing models to complete complex tasks. We add a more comprehensive justification for the selection of specific tools.
> 5. We understand the importance of discussing the potential impacts and implications of stereotype detection in practice. In the revised paper, we provide a more thorough discussion of the unintended consequences of bias mitigation strategies, the role of human judgment in determining stereotypes, and the balance between freedom of expression and risk mitigation.
>
> 6. In response to the concern about providing explicit recommendations or strategies for mitigating biases, we include a section discussing potential bias mitigation approaches and their implications in the revised paper.
>
> **Questions:**
>
> 1. Language agents for stereotype detection in text-to-image models may face limitations in accurately detecting subtle stereotypes or those embedded in complex contexts. Additionally, they may struggle with cases where stereotypes are expressed using sarcasm or irony. We discuss these challenges and potential solutions in the revised paper.
> 2. For manual annotation, annotators were provided with guidelines that included descriptions and examples of stereotypes. They were instructed to determine the presence of stereotypes based on these guidelines. We provide more information about the criteria used for manual annotation in the revised paper.
> 3. We apologize for the lack of detail regarding the annotation process. In our study, multiple annotators were involved, and we conducted an inter-rater reliability assessment to ensure consistency. We include this information in the revised paper.
> 4. The toxic text datasets used to construct the benchmark dataset were selected based on their relevance to the task, diversity of content, and representation of various social platforms. We provide more details about the selection criteria and guidelines in the revised paper.
>
>
>
> [1] Jaemin Cho, Abhay Zala, and Mohit Bansal. 2022. DALL-Eval: Probing the Reasoning Skills and Social Biases of Text-to-Image Generative Models. https: //doi.org/10.48550/ARXIV.2202.04053.
>
> [2] Federico Bianchi, Pratyusha Kalluri, Esin Durmus, Faisal Ladhak, Myra Cheng, Debora Nozza, Tatsunori Hashimoto, Dan Jurafsky, James Zou, and Aylin Caliskan. 2022. Easily Accessible Text-to-Image Generation Amplifies Demographic Stereotypes at Large Scale. https://doi.org/10.48550/ARXIV.2211.03759
>
> [3] Naik R, Nushi B. Social Biases through the Text-to-Image Generation Lens[J]. arXiv preprint arXiv:2304.06034, 2023.

---

> > ### Comment · Reviewer_35MJ · 2023-11-22
> > **Thanks.**
> >
> > Thank you for addressing the concerns raised in my review. While I appreciate the additional clarifications and proposed revisions, my overall assessment remains unchanged. I encourage you to continue refining your approach, particularly in terms of capturing a more diverse and representative set of stereotypes and providing a more detailed justification for the selection of specific tools and methodologies. This work has the potential to significantly contribute to the field, and I look forward to seeing its evolution.

---

### Official Review · Reviewer_d3He · 2023-11-01

**Soundness:** 3 good
**Presentation:** 4 excellent
**Contribution:** 2 fair
**Rating:** 6
**Confidence:** 3

**Summary:**

This paper introduces an orchestration of LLM-based tools to evaluate and assess bias in a several text-to-image models. The agent framework takes a text given as input and interprets the query in terms of specific instructions in terms of paired prompts and subgroups. These are then formatted into an optimized prompt for the text-to-image model. A stereotype score is then calculated based on the model output.
This framework is then used to compare a range of popular models, some of which, such as chilloutmix displaying high stereotype scores according to their model. Finally a comparison with human labels is performed, to show the robustness of the scoring framework.

**Strengths:**

This is an interesting study that brings novel ideas for how to systematically assess text-to-image models for stereotypical biases. The orchestration of an agent framework makes this model modular so that it can easily be applied to new models, and can easily be extended to include further stereotypes or benchmarks that might be of interest. As the prevalence of AI generated content increases with the wider adoption of such models, understanding their biases and being able to quickly assess new models and releases is of high topical interest in AI safety and fairness.

**Weaknesses:**

The technical novelties of this paper are quite limited, as it is an automated assessment framework for stereotypes in text-to-image models. The models as well as the metrics considered are from the existing literature.

**Questions:**

While it is included in the submission auxiliary materials, there is no mention of open sourcing the code in the paper.
In my opinion this study should only be accepted if the code is included in an easy to access format, such that the study performed in this paper can easily be reproduced by other researchers on new models.

---

> ### Author Response · Authors · 2023-11-21
>
> Regarding the technical novelty of our paper, we understand your point about the use of existing models and metrics. However, our primary contribution lies not in the novelty of the models or metrics, but in the innovative application of these tools. Our core motivation is to build a language agent that can perform task planning and effectively utilize various tools to detect and identify a broader range of stereotypes present in social media texts. Therefore, this agent overcomes the limitations of previous work that required exhaustive testing of all possible occupations and other descriptors to construct prompts.
>
> We appreciate your concerns regarding open sourcing the code of our study. While we have included the code in the submission auxiliary materials, we understand the need for it to be easily accessible. Therefore, we commit to open sourcing the code in an easy-to-access format in subsequent papers. We will also provide ample examples to illustrate the seriousness of stereotypes in existing text-to-image models, further aiding in the understanding and reproduction of our work.

---

### Official Review · Reviewer_xoCn · 2023-11-03

**Soundness:** 2 fair
**Presentation:** 1 poor
**Contribution:** 2 fair
**Rating:** 5
**Confidence:** 3

**Summary:**

In this paper the authors build a system on top of an LLM to generate images and detect stereotypes in text to image models.  They use the LLM to generate pairs of groups and stereotypes, and then to run classification with a tool on the generated images.  They demonstrate that across multiple models there are significant stereotypes in the generated images.

**Strengths:**

S1. Detecting stereotyping in text-to-image models is important

S2. Using LLMs to drive stereotype detection is a good idea, and the idea of taking this to higher levels of abstraction for an agent is intriguing.

S3. The approach does seem effective in uncovering stereotypes.

**Weaknesses:**

aper seems to try to do too many things and as a result I believe doing none of them sufficiently well and adding confusion to the paper:

W1a. The paper is framed around the method proposing an autonomous agent for stereotype detection. This is a great vision, but the method seems to (a) follow a consistent, pre-determined sequence of actions for the task, and (b) it seems far from autonomous in relying on human intervention and a lot of custom steps (unique datasets, custom prompts, human feedback, etc) at every stage.  This to me is not a critique of the method but that it shouldn't be over-complicated or over-sold as an autonomous agent rather than a reliable process for red-teaming for stereotypes building on LLMs as a tool in that process.

W2a. There has been a fair amount of work on sterotype detection which this work is not compared to and does not grapple with similar issues.  For example, what if there are multiple people in the same image?  How is the diversity and coverage of the generated concerns? When is this better than a curated list? For example, this claim is good but I'd like to experimental evidence: "However, this approach to bias evaluation has its limitations, as it often neglects the more subtle stereotypes prevalent in everyday expressions. These biases frequently manifest in toxic content disseminated across various social platforms, including racial slurs"

W3a. [less critical] The method is framed as benchmarking but I think is better explained as automated red-teaming.  Because the metrics and distribution is less controlled, understanding this as a consistent benchmark seems challenging but the discovered issues are still important as in red-teaming.

W2. Related to the above point, it is hard to gauge how diverse the stereotypes uncovered are and how diverse the images are.  (Are all imges generated with "The people who ___")

**Questions:**

I'd love to see greater understanding of the diversity of stereotypes, a comparison with past work, and more clarity on the autonomy and flexibility of the agent to new tasks.

**Details Of Ethics Concerns:**

The paper is targeting a good, ethical application (detecting stereotypes) but given the level of automation, I'd like to see an ethics review to ensure the relevant concerns are being handled with sufficient sensitivity.

---

> ### Author Response · Authors · 2023-11-21
>
> 1. Weaknesses:
>
> W1a: We acknowledge your concerns regarding the framing of our method as an autonomous agent and agree that it might have been overly ambitious. While you pointed out the dependence on "human intervention," it is crucial to note that, at the present stage, crafting well-written prompts manually remains a vital component in enhancing the performance of an agent. Our intention was to highlight the potential for autonomy rather than present a fully autonomous system. In light of your comments, we will revise our framing to better represent the current state of the method and potential future developments.
>
> W2a: We compare our proposed agent framework with existing stereotype detection methods. It's important to highlight that these conventional methods rely on manually crafted prompts for comprehensive testing.
>
> | Bias dimensions | Ours     | Cho et al. [1] | Bianchi et al. [2] | Naik et al. [3] |
> | --------------- | -------- | -------------- | ------------------ | --------------- |
> | Gender          | &#x2714; | &#x2714;       | &#x2714;           | &#x2714;        |
> | Race            | &#x2714; | &#x2714;       | &#x2714;           | &#x2714;        |
> | Religion        | &#x2714; | &#x2716;       | &#x2716;           | &#x2716;        |
>
> | Bias subjects     | Ours     | Cho et al. [1] | Bianchi et al. [2] | Naik et al. [3] |
> | ----------------- | -------- | -------------- | ------------------ | --------------- |
> | Person            | &#x2714; | &#x2714;       | &#x2716;           | &#x2714;        |
> | Occupacations     | &#x2714; | &#x2714;       | &#x2714;           | &#x2714;        |
> | Traits            | &#x2714; | &#x2716;       | &#x2714;           | &#x2714;        |
> | Situations        | &#x2714; | &#x2716;       | &#x2716;           | &#x2714;        |
> | Verb-noun prompts | &#x2714; | &#x2716;       | &#x2716;           | &#x2716;        |
>
> W3a: We agree with your observation that the method is better explained as automated red-teaming rather than benchmarking. We will revise our framing accordingly to ensure that the discovered issues are highlighted as important aspects of red-teaming, rather than focusing on consistent benchmarking.
>
> 1. Questions:
> In response to your questions, we would like to provide the following information:
> - Diversity of stereotypes: Our experiments uncovered more than 2800 prompts that simultaneously showed stereotypes pointing to one subgroup across multiple models. These prompts cover the special expression of "verb + noun" that was missing in previous studies, such as "commit sadistic murders" and so on.
> - Autonomy and flexibility of the agent: We acknowledge that our method currently involves human intervention and custom steps. However, our goal is to progressively refine these processes towards greater autonomy and flexibility for new tasks.
>
> Regarding the diversity of images, we would like to clarify that all prompts start with a fixed "The people who ___" to avoid the prefix prompt introducing additional stereotypes. This approach ensures that the uncovered stereotypes are solely attributed to the models and not influenced by the prompts themselves.
>
>
> [1] Jaemin Cho, Abhay Zala, and Mohit Bansal. 2022. DALL-Eval: Probing the Reasoning Skills and Social Biases of Text-to-Image Generative Models. https: //doi.org/10.48550/ARXIV.2202.04053.
>
> [2] Federico Bianchi, Pratyusha Kalluri, Esin Durmus, Faisal Ladhak, Myra Cheng, Debora Nozza, Tatsunori Hashimoto, Dan Jurafsky, James Zou, and Aylin Caliskan. 2022. Easily Accessible Text-to-Image Generation Amplifies Demographic Stereotypes at Large Scale. https://doi.org/10.48550/ARXIV.2211.03759.
>
> [3] Naik R, Nushi B. Social Biases through the Text-to-Image Generation Lens[J]. arXiv preprint arXiv:2304.06034, 2023.

---

> > ### Comment · Reviewer_xoCn · 2023-11-22
> > **Thanks!**
> >
> > Thank you for the considerate response!  I hope that the framing changes end up improving the paper.  With respect to W2a, it'd be nice if there were empirical comparisons with those papers at least on the domains they cover.  Did I miss this in the paper?
> >
> > Overall - I still appreciate the paper but will keep my overall assessment. Thanks!

---

### Meta-Review · Area_Chair_Cv4i · 2023-12-05

**Metareview:**

The paper presents a language agent architecture designed for detecting implicit stereotypes in text-to-image models. This approach is significant for its attempt to address ethical risks and biases in AI-generated content, particularly in the context of societal stereotypes like gender, race, and religion. The agent, which is capable of generating instructions and detecting stereotypes in generated images, uses a benchmark dataset constructed from toxic text datasets. However, the paper's strengths in introducing a new approach and addressing timely issues are somewhat overshadowed by its weaknesses. These include a lack of comprehensive evaluation of the agent framework, failure to compare it with existing stereotype detection methods, and an imbalance in the diversity of the benchmark dataset. Additionally, the paper does not delve deeply into the potential impacts of stereotype detection, missing discussions on the optimization of selected tools and strategies for bias mitigation. While the agent aligns well with manual annotation in detecting stereotypes, the paper's overall contribution is somewhat diminished by these shortcomings, highlighting a need for further refinement and broader consideration of the complexities involved in ethical AI development.

**Justification For Why Not Higher Score:**

Not enough support for acceptance from reviewers.

**Justification For Why Not Lower Score:**

NA

---

### Decision · Program_Chairs · 2024-01-16

Reject